# Evaluation Method for Resin Mold Using Reflective Wavefront Sensor

**DOI:** 10.3390/s25216682

**Published:** 2025-11-01

**Authors:** Kazumasa Tatsumi, Kentaro Saeki, Shin Kubota, Yoshikatsu Kaneda, Kenji Uno, Kazuhiko Ohnuma, Tatsuo Shiina

**Affiliations:** 1SEED Co., Ltd., 2-40-2 Hongo, Bunkyo-ku, Tokyo 113-8402, Japan; k_saeki@seed.co.jp (K.S.); shin_kubota@seed.co.jp (S.K.); yoshikatsu_kaneda@seed.co.jp (Y.K.); kenji_uno@seed.co.jp (K.U.); kazuhiko.oonuma625@gmail.com (K.O.); 2Graduate School of Science and Engineering, Chiba University, 1-33 Yayoi-cho, Inage-ku, Chiba-shi 263-8522, Chiba, Japan; shiina@faculty.chiba-u.jp; 3Laboratorio de Lente Verde, 98-1 Nozomino, Sodegaura 299-0251, Chiba, Japan

**Keywords:** wavefront sensor, OCT, contact lenses, resin mold, injection molding

## Abstract

Recent advances in molding technology have enabled the fabrication of plastic molded components with complex geometries. In contact lens (CL) manufacturing, a double-sided molding process using resin molds is employed, in which the front and back surfaces of the lens are replicated through injection molding. However, thermal deformation during polymerization can alter the mold shape, thereby affecting the optical characteristics of the final lenses. This study proposes a high-precision optical evaluation method for resin molds used in contact lens (CL) manufacturing, utilizing a reflective wavefront sensor and optical coherence tomography (OCT). The wavefront sensor demonstrated high measurement accuracy (≈1/100λ) and reproducibility (≈1/200λ) as confirmed using reference samples, and yielded values of approximately 0.012–0.015 μm for the resin molds. Five mold designs with radii of curvature ranging from 6.500 to 8.500 mm were evaluated, revealing that Zernike coefficients varied depending on design and thermal treatment conditions. In particular, astigmatism (Z04) and coma aberrations (Z07) exhibited pronounced trends. A strong correlation was also observed between the Zernike coefficient Z07 and the mold thickness asymmetry measured by OCT. When the thickness difference increased by 2.3 times due to thermal treatment, Z07 increased to 1.9 times. In contrast, Z04 showed no consistent trend and exhibited significant variability (standard deviation > 0.5 μm) after polymerization. The proposed method enables precise detection of subtle shape variations and aberrations, providing valuable feedback for optimizing molding conditions and improving the quality of contact lens production. Furthermore, this method can also be applied to the quality evaluation of other optical components.

## 1. Introduction

Injection molding technology has rapidly advanced in recent years and is now one of the most critical technologies in the plastics industry. In 1872, John W. Hyatt invented the vertical celluloid molding machine, establishing the foundation of injection molding. Since then, this technology has significantly developed, particularly in the latter half of the 20th century [1]. Progress in injection molding techniques and improvements in plastic materials have led to remarkable increases in production efficiency. Furthermore, implementing automated control systems has enabled the fabrication of increasingly complex and high-precision components [2]. Since the 1990s, injection molding has been extensively utilized across various fields, including medical devices and precision optics.

In recent years, with the widespread use of small display devices such as smartphones, many people have increasingly spent long hours viewing these screens daily. This has led to a rise in issues such as myopia and astigmatism, increasing the demand for vision correction. Consequently, the markets for eyeglasses and contact lenses (CLs) have expanded, driving the need for technologies capable of mass-producing high-quality lenses [3,4]. The concept of CLs was first proposed by Leonardo da Vinci in 1508, and in 1888, Adolf Fick developed glass scleral lenses [5]. Subsequently, there was a transition from glass to plastic lenses, and in the 1960s, Otto Wichterle and Drahoslav Lim developed soft CLs made from hydrogel, which are still widely used today [6].

Conventional glass lenses were manually polished; however, the introduction of injection molding technology enabled the mass production of soft CLs [7]. Plastic soft CLs are manufactured using a molding process that employs molds created by injection molding [8]. In this manufacturing process, CL material is injected into resin molds produced using metal molds, and the parts are assembled to form the CL. Therefore, the final shape of the lens heavily depends on the accuracy of these resin molds. The precision, smoothness, and curve texture of the resin mold surfaces directly affect the optical aberrations of the lenses, and even slight imperfections on the surface can be transferred to the lenses.

For CLs in their final form, methods have been proposed to evaluate the shape and thickness of both front and back surfaces using OCT technology [9,10]. However, previous studies have focused only on measuring the CLs themselves, and evaluation of the thickness and shape of the resin molds used in lens fabrication has not been conducted. Furthermore, while past research has mainly addressed optimization of molding conditions [11,12], quantitative assessment of the influence of mold wavefront characteristics on the final lenses remains insufficient. Since the CL manufacturing process involves polymerization under heat and pressure, there are concerns that thermal effects may cause minute deformation. Clarifying these effects is essential for improving the manufacturing process [13]. Laser interferometers are widely used instruments for evaluating resin molds; however, interpreting interference fringes in interferometric measurements is critical and often involves subjective judgment by the operator, making precise quantitative evaluation difficult [14]. Therefore, this study aims to clarify the optical influence of the resin molds’ manufacturing on the CLs. To achieve this research, a reflective wavefront sensor specialized for measuring the wavefront shape of resin molds was developed and employed to evaluate the molds. Wavefront sensing technology enables high-precision measurement of surface geometry and optical properties of lenses [15]. Wavefront sensing technology was first put into practical use in the 1960s as the Shack–Hartmann method and was initially employed mainly in the fields of astronomy and laser optics. Subsequently, advancements in microlens array fabrication, high-resolution imaging sensors, and digital signal processing have enabled highly sensitive and high-speed wavefront measurements [16]. Today, various types of wavefront sensors—such as interferometry [17], Shack–Hartmann wavefront sensors [18], and pyramid wavefront sensors [19,20]—have been developed, each offering distinct advantages in terms of dynamic range and sensitivity. Among them, the Shack–Hartmann sensor is the most widely used due to its robustness, compact configuration, and applicability to both transmission and reflection measurements. Depending on the optical configuration and detector specifications, it can achieve measurement accuracy on the order of a few nanometers root-mean-square (RMS) within milliseconds, allowing real-time evaluation of optical surfaces. In recent years, new analytical approaches incorporating machine learning and digital holography have been proposed, overcoming previous limitations related to measurement range and system dependency [16]. As a result of these advancements, wavefront sensors have found widespread applications in various fields, including bio-optics, optical communications, and precision molding. Using Zernike polynomials enables wavefront analysis with nanometer-level precision and allows real-time acquisition of aberration data within a few milliseconds [21]. Previous studies have demonstrated that such measurements indicate for evaluating aspheric lenses, which supports the adoption of this method in the present work [22]. In addition, the thickness of the resin molds was measured using OCT measurement system specialized for resin mold inspection [9,10]. The resin molds were comprehensively evaluated based on wavefront measurements and the thickness data. Furthermore, CLs were fabricated using the evaluated resin molds, and the influence of the mold properties on the resulting lenses was assessed and discussed.

Through this study, we aimed to quantitatively understand resin molds’ surface shape and optical aberrations and contribute to the establishment of a higher-precision lens manufacturing process.

## 2. Injection Molding Technology and CL Manufacturing

In manufacturing CLs, the double-mold method is a widely adopted and representative technique. This method utilizes a pair of opposing resin molds to form the anterior and posterior surfaces of the lens [23]. Figure 1 illustrates the general process of the double-mold method. First, high-precision resin molds are fabricated by injection molding using a metal mold (Figure 1a). Next, the lens material is dispensed into the concave side of one resin mold (Figure 1b), and the other mold is then assembled with it to form the lens shape (Figure 1c). Polymerization and curing are subsequently performed using ultraviolet light or heat (Figure 1d), and finally, the resin molds are separated to obtain the CL (Figure 1e) [24].

The final shape and optical properties of the CLs are strongly dependent on the precision of the resin molds employed in the manufacturing process; therefore, the evaluation method for these molds is critical. Laser interferometers and transmission-type wavefront sensors are primarily used to evaluate resin molds and lenses, respectively. The laser interferometer is mainly employed for assessing the surface profile of resin molds, whereas the transmission-type wavefront sensor is used to determine CL optical performance [25,26].

Although laser interferometers can achieve high-precision surface measurements at the nanometer scale, several challenges have been reported. First, operator-dependent factors such as fringe contrast adjustment and phase shift calibration can influence the measurement results, introducing subjectivity into the evaluation [27]. Second, depending on the shape of the sample, particularly in cases of high curvature or aspheric surfaces, the recognition and analysis of interference fringes become difficult, resulting in a limited measurement range and reduced accuracy [28]. For high-curvature samples, the optical path difference changes rapidly, causing the interference fringes to become locally very narrow and densely packed. For aspheric samples, the curvature varies across the surface, resulting in distorted fringe patterns that are no longer straight. In both automated algorithms and visual inspection, it is particularly difficult to accurately identify the fringes in regions containing high-frequency components. As a result, the measurable range and precision are limited [16].

To overcome these limitations, this study developed a reflective wavefront sensor specialized for measuring resin molds and proposes a novel evaluation method utilizing this system. The wavefront sensor enables high-precision, quantitative evaluation without operator subjectivity by employing aberration analysis based on Zernike polynomials. In addition, since laser interferometers measure reflections from samples along a fixed optical axis, the measurement range varies depending on the curvature radius of the sample. In contrast, the reflective wavefront sensor offers the advantage of a fixed measurement range of φ5 mm, as the mask area can be modified in response to changes in the curvature radius of the sample.

On the other hand, transmission-type wavefront sensors are widely used to evaluate the optical properties of CLs. These sensors measure the wavefront of light transmitted through lenses or other transparent optical elements and are suitable for comprehensive optical performance evaluations [26]. However, this method yields a composite wavefront that includes the effects of both the anterior and posterior surfaces of the lens, as well as its thickness. Therefore, separating and analyzing the contribution of each component is difficult.

In this study, we aimed to individually evaluate the wavefront characteristics of resin molds using a reflective wavefront sensor, enabling detailed analysis of how the mold affects the optical performance of the final lens.

## 3. Experimental Set-Up

### 3.1. Wavefront Measurement

This study developed the reflective wavefront sensor to evaluate the resin molds used to produce CLs (Pulstec Industrial Co., Ltd., Hamamatsu, Japan, LUCAS). The specifications of the wavefront sensor are shown in Table 1. A reflective wavefront sensor is a device that measures the wavefront of light reflected from an object’s surface and is commonly used for evaluating mirrors and reflective optical elements [29]. Unlike laser interferometers, which have traditionally been used to assess resin molds, the reflective wavefront sensor can overcome challenges such as measurement results being influenced by operator skill and the measurement range being affected by the curvature radius of the object. Therefore, it is considered applicable as a new evaluation method for resin molds.

However, it is essential to design an appropriate optical configuration to accurately capture the reflected light, as this significantly affects the accuracy of the measurement. Figure 2 illustrates the configuration of the developed reflective wavefront sensor. A pigtailed laser is used as the light source and is delivered via a single-mode fiber. As the light emerging from the fiber tip diverges, it is expanded and collimated by a collimating lens to illuminate the CL with enough beam size. The light reflected from the sample then passes through a microlens array before reaching the imaging sensor. In this measurement system, the diffracted light from the sample is removed by a pinhole. Moreover, the wavefront of the lens placed after the pinhole conjugates to the back focal plane of the microlens array in the wavefront sensor, ensuring that the system is unaffected by aberrations inherent to the optical set-up. Within the measurement ranges shown in Table 1, both concave and convex surfaces can be analyzed for wavefront characteristics. The measurement range covers the diopter range used in contact lens manufacturing and is sufficient for evaluating resin molds. The range can be modified via the software and is determined based on the radius of curvature of the target sample.

As shown in Figure 3, when the wavefront of the object has no distortion, the captured image will display uniformly spaced focal spots corresponding to the microlens array. Conversely, if the wavefront is distorted due to phase changes introduced by the object, the incident direction of the light rays shifts relative to each microlens, causing the Hartmann spot positions to deviate. From these deviations, the wavefront shape of the incoming light can be obtained [30].

### 3.2. Radius of Curvature Measurement

The reflective wavefront sensor developed in this study is capable of measuring not only wavefronts but also the radius of curvature. The principle of this measurement is illustrated in Figure 4. In this method, the radius of curvature is defined as the distance between the standard wavefront measurement position and the retroreflection position. In this study, the standard wavefront measurement position refers to the optical position where the reflected light from the sample becomes collimated. This position is defined as the point where the Zernike coefficient Z03 (Defocus) is less than 0.100 μm, allowing wavefront measurements to be obtained with minimal influence from optical system aberrations.

The retroreflection position refers to a specific point on the optical axis where the incident light reflects off the object’s surface and returns along the same optical path toward the light source. Since the retroreflection position can be quantitatively determined, this approach eliminates operator-dependent variations problematic in conventional measurement techniques.

In this measurement, the amount of displacement from the standard wavefront measurement position to the retroreflection position was defined as the radius of curvature.

### 3.3. Analysis Using Zernike Polynomials

The positional information of the Hartmann spots, as described in Section 3.1, is expressed using Zernike polynomials, a set of orthogonal polynomials defined on the unit circle. Each Zernike polynomial represents a specific wavefront shape and corresponds to a particular type of aberration.

Zernike polynomials are widely used to represent wavefront aberrations in optical elements with circular apertures. Since each coefficient corresponds to a specific aberration component, they are effective for quantitatively evaluating distortions or deformations in optical product shapes [31].

In wavefront analysis, it is necessary to mathematically decompose the measured wavefront shape to evaluate the aberration components quantitatively. Therefore, Zernike polynomials are commonly employed in wavefront sensor analysis. These polynomials form an orthogonal set of functions defined within a circular region and allow the wavefront to be decomposed into low- to high-order components.

The incident equiphase surface is expressed using Zernike polynomials, as shown in Equation (1) [32]. $W\left(X,Y\right)$ represents the wavefront phase at the position $\left(X,Y\right)$ on the measurement plane. $Z_{i}^{2j-1}\left(X,Y\right)$ denotes the Zernike polynomial corresponding to radial order “*i*” and azimuthal frequency “2j−1”. These functions are orthogonal over a unit circular domain, allowing different types of aberrations to be described independently. The coefficient $c_{i}^{2j-1}$ indicates the weight of each Zernike term and serves as a quantitative indicator of the magnitude of each aberration component.(1)$$W\left(X,Y\right)=\sum_{i=0}^{n}\sum_{j=0}^{i}c_{i}^{2j-1}Z_{i}^{2j-1}\left(X,Y\right)$$

This is example 2 of an equation: As shown in Section 3.1, when a distorted wavefront is measured, the positions of the Hartmann spots are displaced. Denoting this displacement as (∆*x*, ∆*y*) and the focal length of the microlens array as “*f*”, the relationship between the wavefront tilt and the spot displacement is given by Equation (2).(2)$$\frac{\partial W(X,Y)}{\partial X}=\frac{∆x}{f}\;\frac{\partial W(X,Y)}{\partial Y}=\frac{∆y}{f}$$

The Zernike coefficients can be determined by substituting Equation (1) into Equation (2), enabling wavefront reconstruction. In this study, the evaluation of the resin mold was conducted using Zernike coefficients. Although there are 36 Zernike parameters, this research focuses only on the coefficients that showed notable characteristics (see Table 2).

The coefficient “Z04” corresponds to astigmatism. Astigmatism is one of the five Seidel aberrations (spherical aberration, coma, astigmatism, field curvature, and distortion). In an optical system, when orthogonal coordinates are defined as the horizontal and vertical planes, astigmatism refers to the phenomenon where the focal positions of light rays in the horizontal and vertical planes differ. As a result, the image appears as an ellipse, circle, or line, depending on the degree of aberration. In optical lenses, astigmatism arises when the radius of curvature in the horizontal direction differs from that in the vertical direction.

The coefficient “Z07” represents coma aberration. Coma is an optical aberration that causes the image of a point source to spread asymmetrically when the incoming light is off the optical axis. It is caused by the optical system’s asymmetrical shape errors or tilts [33]. Since this aberration depends on the distance from the optical axis, it significantly affects the perceived image quality when viewing through a lens.

These aberrations—astigmatism and coma—are significant in manufacturing CLs. The manufacturing process must be designed to minimize or eliminate these aberrant components. If such aberrations are present in the resin molds, it is possible to analyze and clarify their correlation with the aberrations observed in the final CLs.

## 4. Material and Method

### 4.1. Measurement Sample

In this study, we measured and evaluated the wavefront and radius of curvature in the optical zone of resin molds used for CL production. We subsequently assessed the lenses fabricated using those molds.

Figure 5 shows the design of the CLs created in this study. The Optical Zone (OZ) is the region precisely engineered for vision correction and is typically shaped by altering the front surface of the lens, which does not come into contact with the eye [34].

Figure 6 illustrates the fitting state of the double-mold process used in lens fabrication. Since the front surface design of the CL determines its vision correction power, the optical zone of the Front Curve mold, indicated in orange in Figure 6, is modified to adjust the lens power. In this study, five samples were prepared by varying the radius of curvature in the OZ region of the front-side resin mold, as listed in Table 3. Evaluation was performed on the resin molds and the CLs fabricated from them.

It is important to note that only the curvature radius of the OZ region in the front-side mold was altered in this study. At the same time, all other design parameters were kept identical. Additionally, all the backside molds were fabricated using a uniform design.

### 4.2. Method

Measurements were performed at each production stage to verify changes in the resin mold throughout the CL manufacturing process. First, wavefront measurements were conducted on the metal mold in step Figure 1a. Next, before injecting the material in step Figure 1b, the resin mold’s wavefront and radius of curvature were measured. Then, following the process, CLs were produced. After demolding and removing the lenses in step Figure 1e, the resin mold’s wavefront and radius of curvature were again measured. This approach made it possible to track changes in the resin mold from its initial fabrication through to the completion of the CL.

Subsequently, the produced CLs were measured using a transmission-type wavefront sensor, NIMO (Lambda-X s.a., Nivelles, Belgium, NIMO TR1504) [35,36]. As mentioned in Section 3, the reflective wavefront sensor developed in this study is unsuitable for CL measurement because the lenses are thin films with high transparency, making it very difficult to clearly distinguish reflections from the front and back surfaces. Therefore, NIMO, which is still commonly used, was employed for the CL measurements. The evaluation compared and analyzed the diopter values calculated from the curvature radii measured on the resin molds with the reflective wavefront sensor and those obtained from the combined front and back surface wavefront measured by NIMO.

## 5. Result

To verify the measurement accuracy of the reflective wavefront sensor used in this study, evaluations were conducted using reference samples and resin molds. Specifically, two types of reference samples were employed: a convex glass surface with a radius of curvature of 6.200 mm and a concave glass surface with a radius of curvature of 9.500 mm. In addition, a concave resin mold with a radius of curvature of 6.500 mm was used to assess measurement accuracy. Using these reference samples and resin molds, the average, standard deviation, and RMS values were calculated to evaluate the sensor’s performance. The results are summarized in Table 4. Key Zernike coefficients (Z04 and Z07) as well as the radius of curvature were assessed, confirming that the device provides highly precise (≈1/100λ) and highly repeatable (≈1/200λ) measurements.

### 5.1. Variations in Zernike Coefficients

In this study, to evaluate changes in resin molds during the manufacturing process of CLs, the wavefront shapes of the metal mold (Figure 1a), the molded resin mold (Figure 1b), and the resin mold after thermal treatment (Figure 1d) were measured using a reflective wavefront sensor.

Figure 7 shows the results of wavefront measurements at each stage, with the Zernike coefficient index on the horizontal axis and the absolute values of the coefficients on the vertical axis. First, Figure 7a presents the wavefront measurement of the metal mold. The metal mold was made of aluminum, with a reflectivity of approximately 90–95% at the laser wavelength of 589 nm. By appropriately adjusting the laser power, the Hartmann spots on the sensor were sufficiently bright, allowing the reflected light to be accurately detected by the wavefront sensor. The primary Zernike coefficients were close to zero, and no significant distortion was observed. This indicates that the mold had high shape accuracy, and no notable aberrations were present during fabrication. Next, Figure 7b shows the wavefront of the resin mold formed from the metal mold. Distortions not observed in the metal mold became apparent, and specific trends were found in the Zernike coefficients. In particular, significant changes were observed in multiple coefficients, especially spherical aberration (Z04) and coma (Z07). These coefficients exhibited trends dependent on the designed radius of curvature, suggesting that aberrations corresponding to the design were transferred during the molding process. The occurrence of optical aberrations is closely related to the molding conditions and the shrinkage behavior of the resin, indicating that these factors affect the final shape [37]. Furthermore, Figure 7c shows the wavefront after polymerization. The Zernike coefficients increased overall, and greater distortions were observed. While Z07 increased with the same trend as before polymerization, Z04 showed no such trend and changed in a different direction than the molded state. These results reveal that thermal deformation occurs at each stage of the CL manufacturing process.

### 5.2. Radius of Curvature Measurements

Table 5 presents the radius of curvature measured using the reflective wavefront sensor. For comparison, the same samples were also evaluated using a laser interferometer. In all cases, the measured values were smaller than the design specifications. The results obtained by both methods were consistent within the measurement error range, indicating that the wavefront sensor provides measurement accuracy comparable to that of the conventional laser interferometry technique.

Figure 8 shows the changes in radius of curvature before and after polymerization. The horizontal axis represents the sample number, and the vertical axis indicates the measured radius of curvature. As in the wavefront measurements, the radius of curvature after molding (Figure 1b) and after polymerization (Figure 1d) was compared. In all samples, the radius of curvature decreased after polymerization, confirming that shrinkage occurred during the process.

Table 6 presents the shrinkage ratios of the radius of curvature for each sample. The shrinkage ratio varied depending on the mold design; samples with a larger designed radius of curvature tended to exhibit greater shrinkage. It is generally known that shrinkage in resin molds increases when the radius of curvature is small or when the resin thickness is large [38,39]. Sample (A) exhibited the highest curvature of the samples used in this study, but had the smallest resin thickness. In contrast, Sample (E) had the lowest curvature and the most significant resin thickness. In the present results, Sample (E) showed the highest shrinkage ratio, suggesting that resin thickness is a dominant factor influencing shrinkage rather than curvature.

### 5.3. Results of CL Power Measurement

The evaluation of the CLs was conducted by comparing the lens power measured using NIMO with the power calculated from the radii of curvature of the resin mold, which were obtained using the reflective wavefront sensor. The theoretical lens power of the final CL product was calculated based on the radii of curvature of the anterior and posterior surfaces of the resin mold measured by the reflective wavefront sensor. Equation (3), a standard formula used to calculate the adequate power of a thin optical system with a central thickness, was employed for this purpose [33,40]. In Equation (3), the variables are defined as follows: “D” represents the optical power, “f” the focal length, “n” the refractive index of the CL, “R_FC_” the radius of curvature of the front surface, “R_BC_” the radius of curvature of the back surface, and “t” the central thickness of the CL.(3)$$D=\frac{1}{f}=\left(n-1\right)\left(\frac{1}{R_{FC}}-\frac{1}{R_{BC}}\right)+\frac{{\left(n-1\right)}^{2}t}{nR_{FC}R_{BC}}$$

The lens powers calculated using Equation (3) are shown by the blue line in Figure 9. In contrast, the results of CL power measured by the transmission-type wavefront sensor, using lenses fabricated from the corresponding resin molds, are indicated by the orange line in Figure 9. The comparison between the two curves revealed that samples with a smaller designed radius of curvature exhibited larger discrepancies between the calculated and measured values. Conversely, samples with a larger designed radius of curvature showed fewer minor differences between calculation and estimated values, demonstrating good agreement.

## 6. Discussion

In this study, samples designed with a smaller radius of curvature exhibited significant discrepancies between the CL power calculated from the radius of curvature measured on the resin molds and the power measured directly on the CLs using NIMO. The wavefront data obtained from the reflective wavefront sensor alone were insufficient to fully explain this phenomenon. Therefore, we hypothesized that differences in the resin thickness of the molds, along with shrinkage and distortion during the molding and polymerization processes, will affect the CL power.

Multiple resin molds with varying thicknesses were fabricated to verify this hypothesis. The resin thickness was controlled by changing the spacer (SP), which determines the mold height. Figure 10 shows an overview of the spacer. For this study, the mold was designed with a spacer configuration, allowing easy adjustment of the resin thickness. By changing the spacer thickness, the height of the convex part of the mold can be altered, thereby theoretically controlling the resin thickness. Three types of spacers—SP 4.9 mm, 5.0 mm, and 5.1 mm—were prepared, enabling resin thickness variation in 0.1 mm increments. The results shown in Section 5 correspond to the resin molds fabricated with the 5.0 mm spacer. For five samples with different designed radius of curvature, three thickness conditions were added, resulting in a total of 15 samples. Wavefront and radius of curvature evaluations were conducted for all samples. The final CLs were measured using NIMO.

To measure the resin mold thickness varied by the spacers, the OCT technology specialized for CLs and resin mold measurement was employed to perform non-destructive and non-contact thickness measurements [9,10]. OCT is an imaging technique based on low-coherence interferometry and is widely used in biomedical measurement fields [41]. Recently, its high resolution and rapid scanning capabilities have also enabled applications in industrial thickness measurements [42]. Since OCT visualizes internal structures without destruction or contact, it is a powerful tool for quality evaluation of optical products.

Taking advantage of OCT’s axial (thickness) resolution, the interference signals of the resin mold’s front and back surfaces were identified from their respective reflection signals, enabling precise thickness acquisition. Considering the expected relationship between thickness non-uniformity and wavefront aberrations, two cross-sectional scans were performed: one along the resin injection direction during molding and the other perpendicular to it.

Here, the measured thickness distribution data of the resin molds are additionally presented, and their relationship with the wavefront aberration data is discussed.

### 6.1. Wavefront Measurement Results of Resin Molds Fabricated with Different Thicknesses

Figure 11 shows the wavefront measurement results of resin molds with thicknesses varied by spacers. Figure 11a shows the wavefront results for the resin mold fabricated using SP 4.9 mm, with the post-polymerization results shown in Figure 11b. The resin mold fabricated with SP 4.9 mm is the thickest design in the three types of spacers. In these results, Z04 exhibited trends dependent on the design, while Z07 showed uniform values. Since the thicker resin mold appeared to influence the coma aberration, this effect is discussed in the following section.

Figure 11c shows the wavefront results of the resin mold fabricated using SP 5.0 mm, with the post-polymerization results shown in Figure 11d. Although the results in Chapter 5 also use resin molds fabricated with SP 5.0 mm, different sample molds were employed here. While there are some discrepancies in the results, the trends in the Zernike coefficients were consistent. The Zernike coefficients increased after polymerization, and Z07 reflected trends derived from the design.

Figure 11e shows the wavefront results for the resin mold fabricated using SP 5.1 mm, with post-polymerization results shown in Figure 11f. This mold had the thinnest resin thickness in this study. Regarding the Zernike coefficients, this sample showed the smallest coefficients and the least distortion among the three conditions with different SPs. Even after polymerization, Z07 still showed a trend attributable to the design.

The appearance of the Zernike coefficient trends as the resin thickness decreased suggests a correlation between resin thickness and Zernike coefficients.

### 6.2. Relationship Between Resin Mold Thickness and Zernike Coefficients

To investigate the correlation between resin mold thickness and wavefront measurement results, thickness measurements were conducted using an OCT system designed explicitly for evaluating CLs and resin molds. As OCT enables tomographic imaging, cross-sectional measurements were performed in two directions: the direction of the resin injection and the direction perpendicular to it. For each direction, the resin mold was measured from −20 degrees to 20 degrees by tilting the measurement stage in 2° increments, with the center of the mold defined as 0°. The measurements were conducted on the resin molds after polymerization, as shown in Figure 1d.

Figure 12 presents the thickness distribution results obtained along with the injection and its perpendicular directions. The horizontal axis represents the tilt angle, and the vertical axis shows the measured thickness, allowing a comparison of thickness variation in both directions. Figure 12a,b show the thickness distributions of two resin molds fabricated from Sample (A) using different SPs. The orange bar indicates the injection direction, while the blue bar represents its perpendicular direction. In both Figure 12a,b, the blue line (perpendicular direction) demonstrates a uniform thickness from the center to the periphery, whereas the injection direction exhibits noticeable thickness non-uniformity. This non-uniformity was observed regardless of the SP used. Figure 12c,d show the thickness distributions of two resin molds fabricated from Sample (E) using different SPs. Although thickness non-uniformity was still present, it was smaller compared to that in Figure 12a,b. Notably, Figure 12d, corresponding to the resin mold fabricated using SP 5.1 mm, exhibited the most uniform thickness distribution. The surface slope induced by this thickness variation is a primary cause of the asymmetric wavefront deformation observed in Z07. This explanation aligns with the Z07 trends observed in the wavefront measurement results of this study. Furthermore, under the four conditions shown in Figure 12—(a) Sample A—SP 4.9 mm, (b) Sample A—SP 5.1 mm, (c) Sample E—SP 4.9 mm, and (d) Sample E—SP 5.1 mm—a quantitative correlation was observed between the Zernike coefficient Z07 and the thickness difference in the resin mold. Specifically, in condition (a), the average value of Z07 was 1.197 µm, and the thickness difference in the two directions of the resin mold was 21.2 µm. In condition (b), Z07 was 0.697 µm, and the thickness difference was 18.2 µm. After heat treatment, the Z07 value increased by approximately 1.20 times, and the thickness difference increased by about 1.16 times. Similarly, for conditions (c) and (d), the change in Z07 was approximately 1.9 times, and the change in thickness difference was about 2.30 times. These results indicate a strong correlation between the Z07 aberration and the thickness difference, which depends on the resin design (radius of curvature, thickness) and the injection direction. In particular, under conditions with greater asymmetry in resin mold thickness, the Z07 value tended to be higher, confirming that the reflective wavefront sensor accurately captures such subtle deformations. On the other hand, for Z04, the trend was lost after heat treatment. In all samples, the standard deviation exceeded 0.5 µm, indicating increased variability. From these findings, it was confirmed that Z07 is a strongly correlated and effective indicator for evaluating the relationship between shape and aberration in the samples and molding parameters used in this study.

Additionally, for the samples mentioned above, five resin molds were fabricated and measured under each condition to assess the reproducibility and variability of the wavefront measurements. Table 7 presents the average values, standard deviations, and RMS values for the Zernike coefficients Z04 and Z07, as well as the radius of curvature in the optical region. These results indicate that the variability in wavefront characteristics is influenced by both the resin mold thickness and the designed radius of curvature, providing further insights into the stability of the molding process.

### 6.3. Relationship Between Resin Thickness and Lens Power

Figure 13a–c shows the calculated lens power (blue line) obtained by substituting the measured radii of curvature of the resin molds into Equation (3), along with the actual lens power measured using NIMO (orange line). The theoretical values calculated from the mold design parameters are also indicated (gray line). Figure 13a presents the results for lenses fabricated using resin molds formed with a 4.9 mm spacer (SP 4.9 mm), while Figure 13b,c correspond to lenses molded with 5.0 mm (SP 5.0 mm) and 5.1 mm (SP 5.1 mm) spacers, respectively. The blue lines, representing the calculated power based on the measured radii of the resin molds, showed good agreement with the design-based theoretical values (gray line) in Sample (A). In contrast, Sample (E) exhibited higher calculated power values than expected. As discussed in Section 5.2, this discrepancy can be attributed to the polymerization shrinkage of the resin mold; molds designed with larger radii tend to exhibit a higher shrinkage rate, leading to a greater deviation from the theoretical value. The orange lines representing actual lens power by NIMO measurements consistently showed higher power values than those calculated from the design specifications across all samples. The deviation from the design value was more pronounced in samples with smaller radii of curvature. Although this trend was consistently observed across all resin thicknesses, the smallest discrepancy between the calculated and measured lens powers occurred in the thickest mold condition, SP 4.9 mm.

Thicker resin molds are known to be less susceptible to deformation caused by polymerization shrinkage, thereby exhibiting greater dimensional stability during molding [39]. This tendency was also confirmed in the present study, where increased mold thickness led to improved agreement between the predicted and actual lens power values.

## 7. Summary

In this study, we investigated the optical effects of resin molds used in CL manufacturing by evaluating wavefront aberrations with a reflective wavefront sensor and analyzing thickness distribution using OCT. Five types of molds with different designed radii of curvature were examined before and after polymerization, successfully quantifying minute deformations and aberrations that are difficult to detect with conventional interferometers. Zernike polynomial analysis revealed that astigmatism (Z04) and coma aberration (Z07) varied depending on mold design and thickness. After polymerization, Z07 showed correlation with the designed radius, which is thought to be caused by asymmetric shrinkage. OCT imaging confirmed that directional thickness variations exceeding 10 µm between the injection and non-injection sides were strongly associated with coma aberration. On the other hand, wavefront evaluation of the final lenses showed that these coma aberrations were not transferred, indicating minimal impact on optical properties. Furthermore, lens power calculated from mold curvature showed good agreement with measured values, especially for molds with greater thickness. These results suggest that the combination of the reflective wavefront sensor and OCT provides a practical and highly precise method for mold evaluation, serving as a useful tool for design feedback and optimization of molding conditions.

These evaluation results provide valuable feedback for the injection molding process and contribute to optimizing molding conditions. In the double-mold process commonly used for CL manufacturing, this method enables integrated evaluation of both resin molds and final lenses, offering practical benefits such as feedback to the production process and yield improvement. Furthermore, it can be applied to special lenses such as toric lenses and EDOF (Extended Depth of Focus) lenses.

Moreover, this method is not limited to CLs but can be applied broadly to the quality evaluation of various optical components. For example, it can be used for quantitative assessment of optical performance and wavefront aberrations in microlenses, enabling high-precision verification of product characteristics. It is also effective for analyzing distortions on reflective surfaces such as mirrors, allowing highly sensitive detection and evaluation of image distortions caused by minute surface irregularities through wavefront measurement. This makes it useful as a quality control tool for mirror surfaces and as a feedback metric in manufacturing processes.

## Figures and Tables

**Figure 1 sensors-25-06682-f001:**
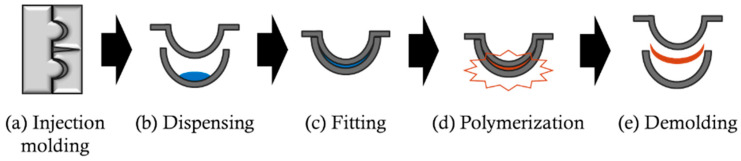
Manufacturing process of CLs using the double-mold method. (**a**) Injection molding process, (**b**) dispensing process, (**c**) fitting process, (**d**) polymerization process, and (**e**) demolding process.

**Figure 2 sensors-25-06682-f002:**
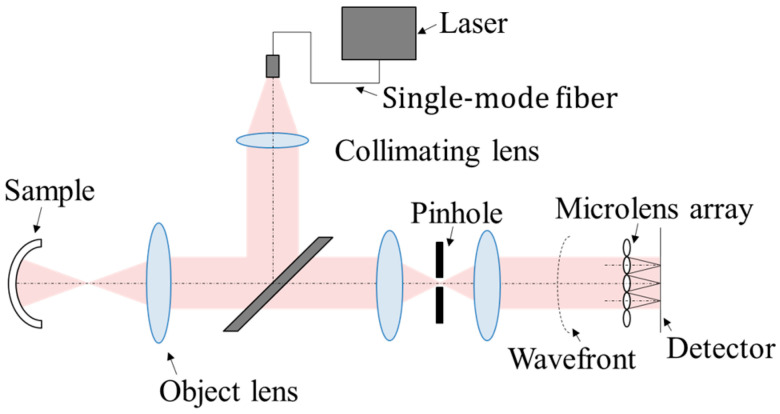
Wavefront sensor principle.

**Figure 3 sensors-25-06682-f003:**
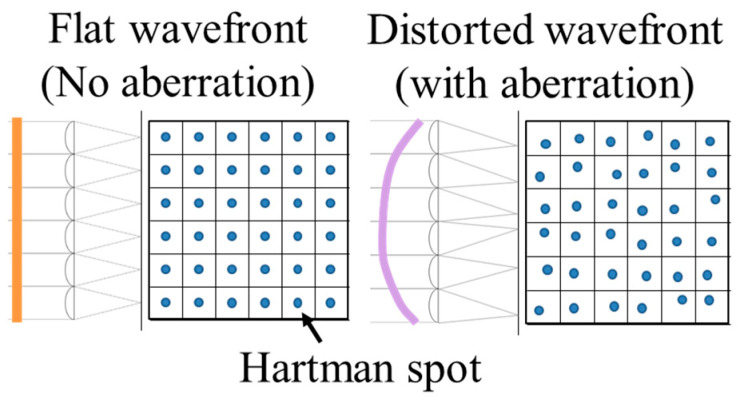
The relationship between wavefront and Hartmann spot.

**Figure 4 sensors-25-06682-f004:**
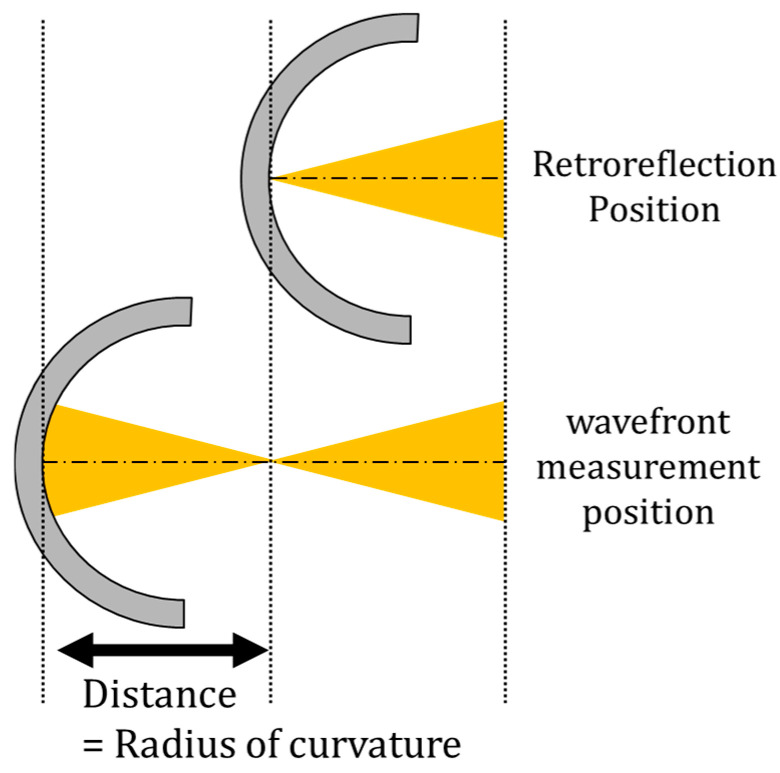
Measurement of radius of curvature.

**Figure 5 sensors-25-06682-f005:**
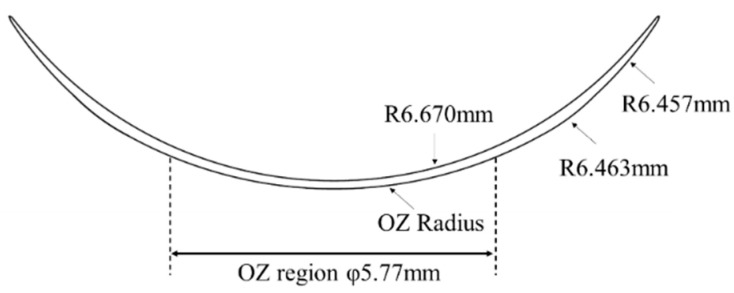
Overview of the CL.

**Figure 6 sensors-25-06682-f006:**
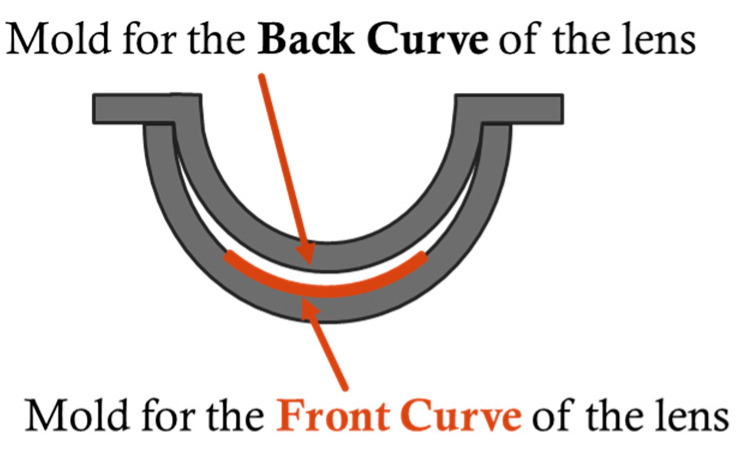
Overview of the resin mold.

**Figure 7 sensors-25-06682-f007:**
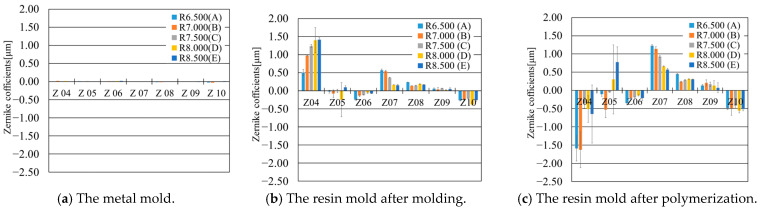
Wavefront measurement results at each processing stage. (**a**) Measurement result of the metal mold, (**b**) measurement result of the resin mold after molding, (**c**) measurement result of the resin mold after polymerization.

**Figure 8 sensors-25-06682-f008:**
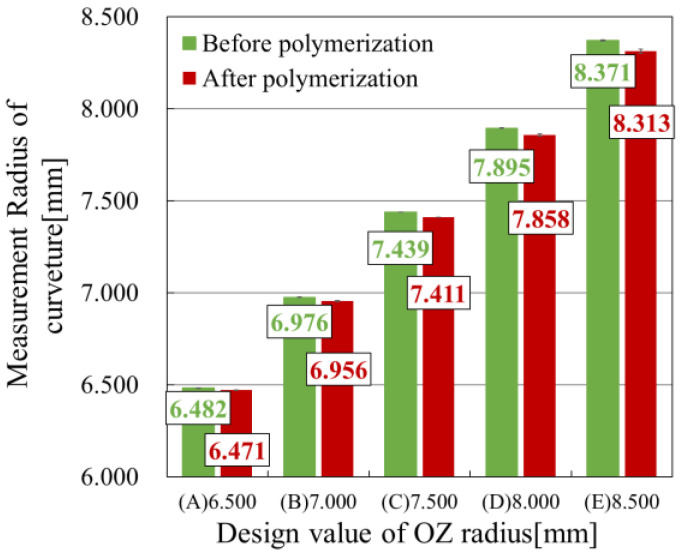
Changes in the radius of curvature of resin molds before and after polymerization.

**Figure 9 sensors-25-06682-f009:**
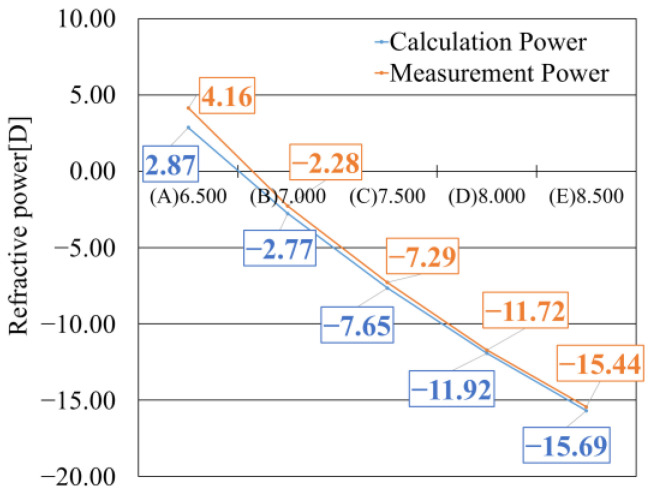
Comparison between calculated and measured CL powers.

**Figure 10 sensors-25-06682-f010:**
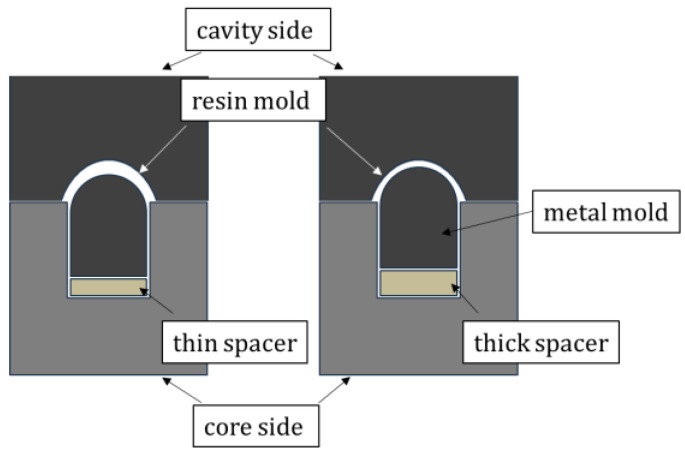
Overview of the spacer arrangement in injection molding.

**Figure 11 sensors-25-06682-f011:**
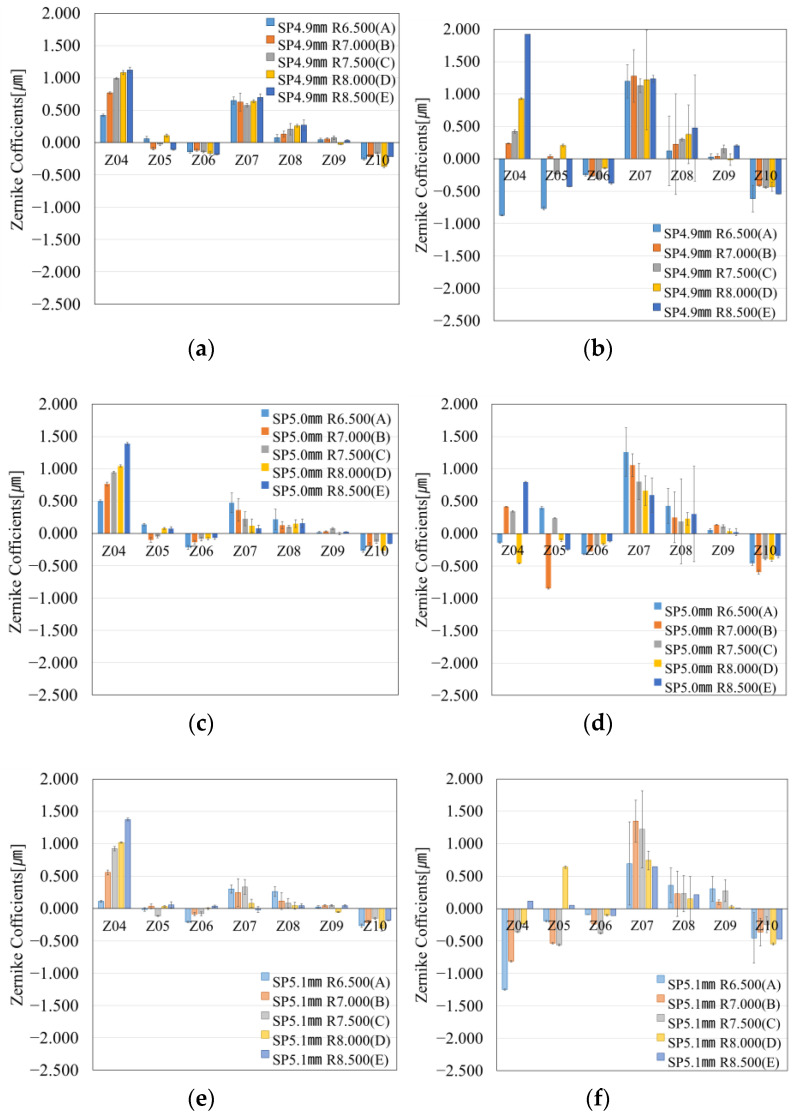
Wavefront measurement results of resin molds fabricated with different SP before and after polymerization. (**a**) Before polymerization, SP 4.9 mm. (**b**) After polymerization, SP 4.9 mm (corresponding to (**a**)). (**c**) Before polymerization, SP 5.0 mm. (**d**) After polymerization, SP 5.0 mm (corresponding to (**c**)). (**e**) Before polymerization, SP 5.1 mm. (**f**) After polymerization, SP 5.1 mm (corresponding to (**e**)).

**Figure 12 sensors-25-06682-f012:**
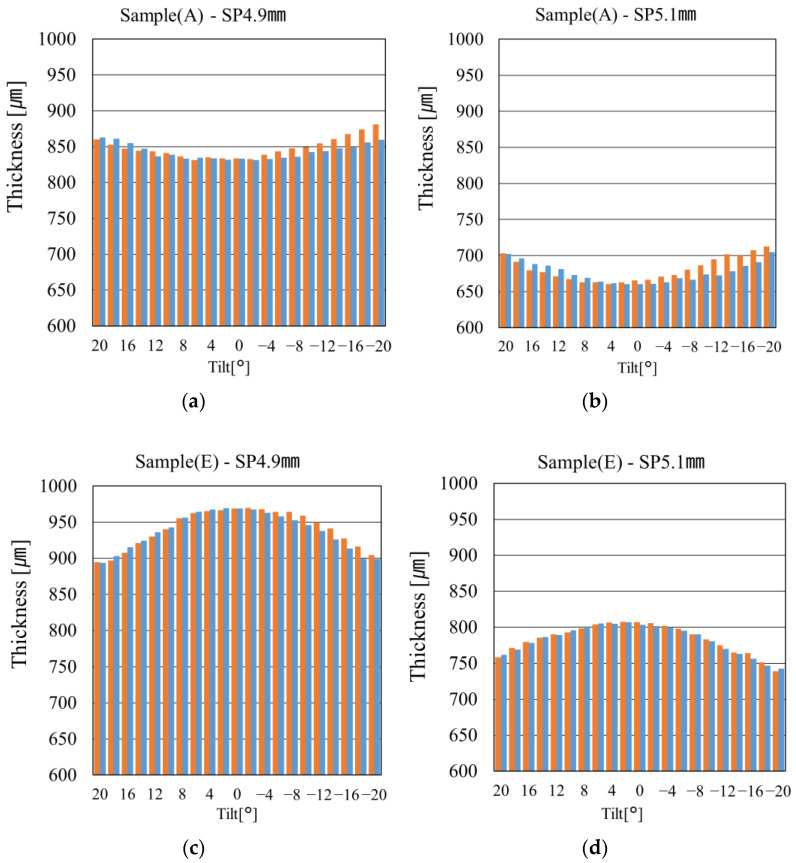
Thickness measurement results of resin molds in the injection and perpendicular directions. The orange bar indicates the injection direction, while the blue bar indicates the perpendicular direction. (**a**) Sample (A)—SP 4.9 mm, (**b**) Sample (A)—SP 5.1 mm, (**c**) Sample (E)—SP 4.9 mm, (**d**) Sample (E)—SP 5.1 mm.

**Figure 13 sensors-25-06682-f013:**
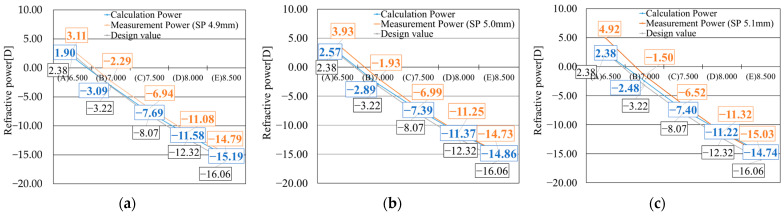
Comparison of refractive power of lenses fabricated using different spacer thicknesses. (**a**) SP 4.9 mm, (**b**) SP 5.0 mm, (**c**) SP 5.1 mm.

**Table 1 sensors-25-06682-t001:** Specifications of the wavefront sensor specialized for resin mold measurement.

Index	Parameter
Manufacturer	Pulstec Industrial Co., Ltd.
Model	LUCUS
Measurement wavelength	589 nm
Measurable radius of curvature for concave surfaces (measurement range)	4.3 mm–46.5 mm (φ7.0 mm)
Measurable radius of curvature for convex surfaces (measurement range)	5.7 mm–9.5 mm (φ9.0 mm)
Tolerance of wavefront incident angle	±1.1 deg
Wavefront measurement accuracy	1/100 λ
Repeatability	1/200 λ
Radius measurement accuracy	±10 μm
Number of microlens array	108 × 80
The focal length of the microlens array	4 mm
Data update rate	6 Hz
Focal lens	NA 0.81

**Table 2 sensors-25-06682-t002:** Zernike polynomials (Fringe index order) and corresponding aberrations.

Term	Polynomial	Aberration
Z01	ρcosθ	x-Tilt
Z02	ρsinθ	y-Tilt
Z03	${2\rho }^{2}-1$	Defocus
Z04	$\rho^{2}cos2\theta$	0° Primary astigmatism
Z05	$\rho^{2}sin2\theta$	45° Primary astigmatism
Z06	${(3\rho }^{2}-2)\rho cos\theta$	Primary x-coma
Z07	${(3\rho }^{2}-2)\rho sin\theta$	Primary y-coma
Z08	${6\rho }^{4}-{6\rho }^{2}+1$	Primary spherical aberration
Z09	$\rho^{3}cos3\theta$	Secondary x-trefoil
Z10	$\rho^{3}sin3\theta$	Secondary y-trefoil

**Table 3 sensors-25-06682-t003:** Specification of resin molds.

Sample No.	Designed Radius of Curvature in the OZ Region [mm]
(A)	6.500
(B)	7.000
(C)	7.500
(D)	8.000
(E)	8.500

**Table 4 sensors-25-06682-t004:** Accuracy Evaluation of the Reflective Wavefront Sensor Using Standard and Resin Mold Samples.

Sample Data			Average Value	Standard Deviation	RMS
Z04	Standard references	convex 6.200 mm	0.023	0.004	0.003
		concave 9.500 mm	0.014	0.005	0.004
	Resin mold	concave 6.500 mm	0.729	0.018	0.015
Z07	Standard references	convex 6.200 mm	−0.044	0.001	0.001
		concave 9.500 mm	−0.093	0.005	0.004
	Resin mold	concave 6.500 mm	0.891	0.015	0.012
OZ radius	Standard references	convex 6.200 mm	6.199	0.001	0.0005
		concave 9.500 mm	9.499	0.001	0.0005
	Resin mold	concave 6.500 mm	6.522	0.001	0.0005

For each parameter, Z04 and Z07 values are expressed in [µm], while OZ radius values are expressed in [mm].

**Table 5 sensors-25-06682-t005:** Comparison of radius of curvature measurements obtained using the reflective wavefront sensor and laser interferometer.

Sample No.	Designed Radius of Curvature in the OZ Region [mm]	Measurement of OZ Radius [mm]	
		Wavefront Sensor	Laser Interferometer
(A)	6.500	6.482	6.476
(B)	7.000	6.976	6.975
(C)	7.500	7.439	7.441
(D)	8.000	7.895	7.899
(E)	8.500	8.371	8.378

**Table 6 sensors-25-06682-t006:** Shrinkage ratio of resin molds during polymerization.

Sample No.	Designed Radius of Curvature in the OZ Region [mm]	Polymerization Shrinkage Rate
(A)	6.500	0.17%
(B)	7.000	0.29%
(C)	7.500	0.38%
(D)	8.000	0.47%
(E)	8.500	0.69%

**Table 7 sensors-25-06682-t007:** Reproducibility evaluation of wavefront measurements for different spacer thicknesses and radius of curvature.

Sample Data			Average Value	Standard Deviation	RMS
Z04	SP 4.9 mm	R6.500	−0.873	0.511	0.417
		R8.500	1.939	0.080	0.065
	SP 5.1 mm	R6.500	−1.740	0.432	0.353
		R8.500	0.122	0.012	0.010
Z07	SP 4.9 mm	R6.500	0.997	0.262	0.214
		R8.500	0.647	0.006	0.005
	SP 5.1 mm	R6.500	1.197	0.413	0.337
		R8.500	1.237	0.004	0.004
OZ radius	SP 4.9 mm	R6.500	6.540	0.005	0.004
		R8.500	8.378	0.003	0.002
	SP 5.1 mm	R6.500	6.500	0.053	0.013
		R8.500	8.314	0.002	0.001

For each parameter, Z04 and Z07 values are expressed in [µm], while OZ radius values are expressed in [mm].

## Data Availability

Data are contained within the article.

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
