# Peer review of "Evaluation Method for Resin Mold Using Reflective Wavefront Sensor"

_sensors, 2025, doi:10.3390/s25216682_

Round 1
Reviewer 1 Report
Comments and Suggestions for Authors
The manuscript titled “Evaluation Method for Resin Mold Using Reflective Wavefront Sensor” presents an intriguing method for evaluating the quality of resin molds used in the fabrication of contact lenses through reflective wavefront sensing combined with optical coherence tomography. In general, the proposed technique shows promise as a complementary approach to existing evaluation methods. However, the manuscript requires several major clarifications and requires significant modifications to improve its clarity and rigor. Please find my detailed comments below:
General Comments:
Line 43: Please provide an appropriate reference to support this statement.
Lines 103-109: Kindly elaborate on why fringe recognition poses challenges in interferometric measurements.
Lines 152-154 and Figure 2: The description of the optical setup contains significant gaps. Please provide a more comprehensive explanation in the main text and ensure the figure is thoroughly annotated. I have several questions regarding the optical configuration:
- In the schematic, why is the laser source depicted as diverging? Although minor, laser sources are generally well-collimated, so clarification is needed.
- To accurately measure the wavefront aberrations introduced by the contact lens surface, this surface should be conjugated to the back focal plane of the microlens array in the wavefront sensor. Please clarify if this is the case.
- The schematic in Figure 2 shows a curved wavefront incident on the microlens array, which contrasts with the expectation of a collimated beam. This discrepancy may affect the uniformity of the Hartmann spot distribution on the sensor. Please describe your sensor calibration methodology and indicate what reference wavefront was used for comparison.
- The positioning of the lens directly in front of the sample is unclear. While displacing the curved contact lens sample can achieve collimation of the outgoing light, this setup may limit the range of radii of curvature that can be measured. Moreover, lenses with larger radii would need to be placed closer to the measurement lens, resulting in measurements over a larger surface area. Conversely, lenses with shorter radii would be placed nearer the focal plane, limiting the sampled surface area. Please discuss these implications.
Line 170: Please clarify what is meant by “standard wavefront measurement.”
Figure 4: This figure is confusing. Please provide a complete schematic of the experimental setup to facilitate reader understanding, including the origin of the laser beam and the reason for the observed focal point.
Table 3: The objective appears to be measuring the radius of curvature using the proposed reflective technique. However, the table references measurements from the interferometer rather than your technique. Please clarify.
Line 265: Please define the acronym “NIMO.”
Figure 7(a): What is the reflectivity of the metal used? What type of metal was employed? Was reflected light detected? Please provide these details.
Lines 319-323: It appears that sample (A) exhibits the lowest curvature while sample (E) has the highest curvature, opposite to what the text suggests. Please address this discrepancy in the text.
Table 4: Only a single numerical column is shown for both the wavefront sensor and the laser interferometer data. Please ensure the data presentation is complete and unambiguous.
Lines 362-363: The role of the spacer is unclear. Please include an illustrative image to explain what is meant by “mold height.”
Figure 11: Please specify the meaning of each color either in the figure caption or by including a legend to aid interpretation.
Author Response
Thank you for your valuable comments. Please see the attachment.

Reviewer 2 Report
Comments and Suggestions for Authors
This study focuses on the development of a method for assessing the quality of polymer molds for contact lens production using a wavefront sensors. The relevance of this work stems from the need to monitor mold deformations at the micron level. The method has significant potential for application in optical manufacturing, but requires some refinement. The ability to correlate process parameters with the optical characteristics of finished products has been demonstrated.
Point 1. The abstract should quantitatively indicate the sensitivity of the proposed method or other metrics.
Point 2. The authors provided a brief comparison of several wavefront measurement methods. This requires improvement based on relevant review papers, such as https://doi.org/10.1016/j.optlastec.2025.113342.
Point 3. The correlation between the Zernike coefficients (Z04, Z07) and process parameters could be clarified, with quantitative relationships presented.
Point 4. The results repeatability on an expanded sample needs to be clarified, including a statistical assessment of variations between identical samples.
Point 5. A numerical estimate (e.g., RMS, MAE, MSE, etc.) should be provided to indicate the accuracy of the Evaluation Method Using Reflective Wavefront Sensor based on the data in Tables 3, 4, etc., and Fig. 11.
The proposed methodology represents a useful tool for quality control in the production of optical components. After addressing these concerns, the article may be recommended for publication.
Major Revision
Author Response

(The authors gave the same response as above.)

Round 2
Reviewer 1 Report
Comments and Suggestions for Authors
Dear Authors,
Thank you for addressing my comments and suggestions. In my view, you have effectively incorporated essential details that enhance the clarity and rigor of your manuscript. I believe the presented technique holds significant potential for application in contact lens manufacturing metrology and recommend publication of this work in its current form.
Reviewer 2 Report
Comments and Suggestions for Authors
The authors responded to all questions and comments.
Accept in present form